# Exploration of Preservation Methods for Utilizing Porcine Fetal-Organ-Derived Cells in Regenerative Medicine Research

**DOI:** 10.3390/cells13030228

**Published:** 2024-01-25

**Authors:** Kenji Matsui, Hidekazu Sekine, Jun Ishikawa, Shin Enosawa, Naoto Matsumoto, Yuka Inage, Yoshitaka Kinoshita, Keita Morimoto, Shutaro Yamamoto, Nagisa Koda, Shuichiro Yamanaka, Takashi Yokoo, Eiji Kobayashi

**Affiliations:** 1Division of Nephrology and Hypertension, Department of Internal Medicine, The Jikei University School of Medicine, Tokyo 105-8461, Japan; 2Institute of Advanced Biomedical Engineering and Science, Tokyo Women’s Medical University, Tokyo 162-0056, Japan; sekine.hidekazu@twmu.ac.jp; 3Division for Advanced Medical Sciences, National Center for Child Health and Development, Tokyo 157-8535, Japanenosawa-s@ncchd.go.jp (S.E.); 4Department of Kidney Regenerative Medicine, The Jikei University School of Medicine, Tokyo 105-8461, Japan; 5Department of Pediatrics, The Jikei University School of Medicine, Tokyo 105-8461, Japan; 6Department of Urology, Graduate School of Medicine, The University of Tokyo, Tokyo 113-8654, Japan; 7Department of Urology, The Jikei University School of Medicine, Tokyo 105-8461, Japan

**Keywords:** fetal pig, cell source, organ preservation, cryopreservation, vitrification, xenotransplantation

## Abstract

Human pluripotent stem cells have been employed in generating organoids, yet their immaturity compared to fetal organs and the limited induction of all constituent cell types remain challenges. Porcine fetal progenitor cells have emerged as promising candidates for co-culturing with human progenitor cells in regeneration and xenotransplantation research. This study focused on identifying proper preservation methods for porcine fetal kidneys, hearts, and livers, aiming to optimize their potential as cell sources. Extracted from fetal microminiature pigs, these organs were dissociated before and after cryopreservation–thawing, with subsequent cell quality evaluations. Kidney cells, dissociated and aggregated after vitrification in a whole-organ form, were successfully differentiated into glomeruli and tubules in vivo. In contrast, freezing hearts and livers before dissociation yielded suboptimal results. Heart cells, frozen after dissociation, exhibited pulsating heart muscle cells similar to non-frozen hearts. As for liver cells, we developed a direct tissue perfusion technique and successfully obtained highly viable liver parenchymal cells. Freezing dissociated liver cells, although inferior to their non-frozen counterparts, maintained the ability for colony formation. The findings of this study provide valuable insights into suitable preservation methods for porcine fetal cells from kidneys, hearts, and livers, contributing to the advancement of regeneration and xenotransplantation research.

## 1. Introduction

Tissue engineering using human pluripotent stem cells has advanced, leading to the creation of various organoids; however, it should be noted that the kidney, heart, and liver organoids are still immature compared to in situ fetal organs [1,2,3]. Furthermore, current technology has not been able to induce all of the constituent cell lineups. For example, the protocol to precisely induce human renal stromal progenitor cells has not yet been developed [4]. From this viewpoint, porcine organs are expected as xenogeneic sources for the components of organoids along with advances in various genetic modifications [5,6]; however, adult organ transplantation still requires strong immunosuppression [7].

To overcome the shortcomings of both approaches, a concept called “xenogeneic regenerative medicine” has been proposed, combining regenerative medicine with xenotransplantation. This approach aims to differentiate human nephrons from nephron progenitor cells (NPCs) within the developmental environments of porcine fetal kidneys, creating chimeric kidneys. Xenogeneic fetal kidneys are promising grafts because they can mitigate hyperacute rejection through vascularization from host vessels, which is problematic in the transplantation of adult tissues [8]. Chimeric kidneys have already been reported in rodent models [9]. With a view to clinical application, we have embarked on producing human–pig chimeric kidneys. Namely, porcine fetal kidney cells may potentially supplement deficient components in human progenitor cells and support the maturation of regenerated human tissues.

It is known that the same cell types from different species form chimeric homogeneous cell clusters [10]. Recently, human and mouse chimeric renal organoids were generated by co-culturing mouse fetal kidney cells with induced human NPCs [11]. While chimeric organoids comprising human and pig progenitor cells have not been reported to date, they could serve as valuable tools for studying the differences in developmental processes and inter-species barriers between humans and pigs to generate mature chimeric organs.

While various porcine stem cells have been established [12], considering the limitations of induced cells, progenitor cells derived from porcine fetuses were considered more preferable. To maximize the utilization of these valuable samples including other organs such as hearts and livers, it is necessary to first establish appropriate cell processing and preservation methods for each organ. In previous studies, pig fetal kidneys exhibited maintained differentiation potency when vitrified for whole-organ cryopreservation [13,14]. However, there have been no reports on other organs.

In this study, we assessed and report the suitable preservation conditions for porcine fetal kidneys, hearts, and livers to obtain progenitor cells: non-frozen, post-dissociation freezing, organ rapid freezing, and organ slow freezing (Figure 1A).

## 2. Materials and Methods

### 2.1. Research Animals

All animal experiments were conducted in accordance with the National Institutes of Health Guidelines for the Care and Use of Laboratory Animals. The animal studies were approved by the animal ethics committee of the Jikei University School of Medicine (approval number: 2023-006). Outbred microminiature pigs (MMPs) originally established from potbelly strain were supplied by Fuji Micra Inc. (Shizuoka, Japan). Male NOD/Shi-scid, IL-2RgKO Jic mice (NOG mice; CLEA Japan, Inc., Tokyo, Japan) were used as recipients for kidney spheroids.

### 2.2. Embryo and Organ Sampling from Pigs with Fixed Gestation Date

A female MMP in the estrus period, judged from the macroscopic changes of external genitalia, was mated with a male MMP for 10–20 min. Mating was performed two times for two consecutive days. Around 25 days after mating, pregnancy was confirmed through abdominal ultrasonography (HS-101V; Honda Electronics Co., Ltd., Aichi, Japan). The first mating day was regarded as the beginning of gestation (day 0). The fetuses were retrieved via cesarean section on embryonic day 30 (E30) under inhalation of isoflurane and intramuscular injection of xylazine. The fetuses were promptly decapitated and transported on ice in 5 mL tubes (30119401; Eppendorf, Hamburg, Germany) containing 4 mL minimum essential medium (MEMα; 12561-056; Invitrogen, Waltham, MA, USA), which had been pre-equilibrated with a 5% CO_2_ atmosphere [15]. Kidneys, hearts, and livers were extracted under a stereomicroscope (M205FA; Leica Microsystems, Wetzlar, Germany) within around 5 h after fetal extraction [14]. Maternal pigs were kept normally after closing the uterus and the abdomen.

### 2.3. Organ Vitrification and Thawing

Cryopreservation through vitrification of kidneys, hearts, and livers was performed as previously reported [13,14]. Because this procedure has been utilized for ovarian tissue slices of 1 mm × 10 mm × 10 mm [16], hearts and livers were sliced into three pieces to facilitate solution penetration. The kidneys and the pieces of hearts and livers were equilibrated in 4 mL base medium (MEM α supplemented with 20% fetal bovine serum (FBS; SH30070.03; HyClone Laboratories, Inc., Logan, UT, USA) and 1% antibiotic–antimycotic solution [15,240,062; Thermo Fisher Scientific, Waltham, MA, USA]) with 7.5% ethylene glycol (EG; 055-00996; FUJIFILM Wako Pure Chemical Corporation, Osaka, Japan) and 7.5% dimethyl sulfoxide (DMSO; 317275-100ML; Millipore, Burlington, MA, USA) on ice for 15 min and soaked in 4 mL base medium with 15% EG and 15% DMSO with 0.5 M sucrose on ice for an additional 15 min. Next, the organs were placed on Cryotops (81111; Kitazato Corporation, Tokyo, Japan) and immersed directly into liquid nitrogen and stored for less than one month. Frozen kidneys and hearts were thawed immediately before use. The Cryotops were quickly transferred from liquid nitrogen to base medium containing 1 M sucrose preheated to 42 °C for 1 min, transferred to base medium with 0.5 M sucrose at room temperature for 3 min, and finally washed twice in base medium at room temperature for 5 min each.

### 2.4. Organ Slow Freezing and Thawing

Some hearts and livers were transferred to cryotubes containing 0.5 mL STEM-CELLBANKER (Zenogen Pharma CO., LTD., Fukushima, Japan) for each fetus, frozen slowly in the −80 °C deep freezer for 1 h, and stored in liquid nitrogen. Shortly before use, cryotubes were placed in a prewarmed water bath at 37 °C. Once thawed, hearts and livers were promptly transferred for dissociation process.

### 2.5. Enzymatic Dissociation of Kidneys and In Vivo Differentiation of Kidney Spheroids

Kidney spheroids were prepared from renal progenitor cells derived from vitrified and thawed porcine fetal kidneys and were transplanted under the left renal capsule of immunocompromised mice, as previously described in mouse fetal kidneys with a slight modification [17]. Kidneys were cut into 4–6 pieces with micro-tweezers (11253-25; Dumont, Montignez, Switzerland), collected into 1.5 mL tube containing 1 mL Accutase (Innovative Cell Technologies, San Diego, CA, USA), and vortexed for 30 s. They were incubated at 37 °C for 15 min, with vortexing after 5 min and gentle manual pipetting using a 200 μL pipette tip after 10 and 15 min, then centrifuged at 300× *g* for 5 min. Pellets were resuspended in 1 mL base medium supplemented with 10 μM Y27632 (257-00511; FUJIFILM Wako Pure Chemical Corporation) and dissociated via gentle manual pipetting. Cell suspensions were passed through a 40 μm cell strainer (BD Falcon, Franklin Lakes, NJ, USA). The suspension density was adjusted to 1 × 10^6^ cells/mL, and 2 × 10^5^ cells were distributed in each well of a 96-well plate (Thermo Scientific, Waltham, MA, USA). Finally, the plate was centrifuged at 1000× *g* rpm for 4 min and incubated overnight at 37 °C to form re-aggregates. The next day, the recipient male NOG mice were anesthetized with isoflurane inhalation and a midline abdominal incision was made in each of them. The intestine was moved to the side to expose the left kidney, and the capsule in the lower part of the kidney was dissected at nearly 1 mm using a microshear. The tip of the outer cylinder of the 22G Surflo I.V. Catheter (Terumo, Tokyo, Japan) was cut at an angle. The outer cylinder was inserted through the incision in the renal capsule with its cut surface facing the renal parenchyma, and a small amount of saline was placed under the renal capsule to detach the renal capsule from the parenchyma. A spheroid was inhaled to the outer cylinder of the Surflo and was inserted under the renal capsule through the incision. The abdomen was then closed with a 5–0 thread. On day 14, recipient mice were euthanized and the transplanted spheroids were collected.

### 2.6. Enzymatic Dissociation and Post-Dissociation Freezing of Hearts

Hearts were dissociated using Multi-Tissue Dissociation Kit and gentleMACSTM Dissociator (Miltenyi Biotec, Bergisch Gladbach, Germany), according to the manufacturer’s protocol. Some cells were suspended in CELLBANKER1 (Zenogen Pharma CO., LTD.) at a density of 1 × 10^6^/mL, transferred to cryotubes, frozen slowly in the −80 °C deep freezer, and stored in liquid nitrogen. Shortly before use, cryotubes were placed in a prewarmed water bath at 37 °C. Once thawed, heart cells were promptly transferred for cell culturing.

### 2.7. Heart Muscle Cell Culture

Non-frozen or cryopreserved heart cells were seeded at a density of 2 × 10^5^ cells per 35 mm dish pre-coated with FBS and cultured for 5 days in MEM α supplemented with 5% fetal bovine serum. On day 4, the dishes were observed under a phase-contrast microscope and the presence of pulsating cells was assessed.

### 2.8. Hepatocyte Dissociation by Direct Tissue Perfusion

Non-frozen or cryopreserved livers were stored tentatively in University of Wisconsin (UW) solution (Belzer UW cold storage solution; Preservation Solutions, Inc., Elkhorn, WI, USA) at 4 °C. Hepatocytes were dissociated with collagenase via perfusion as described previously [18] with the following modifications: (1) livers were not perfused with conditioning buffer and (2) collagenase concentration was set at 10 mg/mL, 10 times the original concentration. A liver was washed with phosphate-buffered saline (PBS) and placed on glass dish warmed on the water bath at 39 °C. Three mL of collagenase solution filled in 5 mL syringe was gently perfused into liver tissue through 27-gauge needle inserted just beneath the serosa of the liver (Figure 2C). Then, digested liver tissue was dispersed gently through pipetting, and we added 20 mL of Williams medium E (W1878; Sigma, Thermo Fisher Scientific, Waltham, MA, USA) containing 10% FBS. Cell suspension was filtrated through nylon mesh with 100 μm spacing (352360; Cell Strainer; Corning, NY, USA) and centrifuged at 190× *g* for 3 min. The resulting precipitates were regarded as comprising the fraction of hepatic parenchymal cells, presumably hepatoblasts in the fetal liver (Figure 2D). The hematopoietic cells and nonparenchymal cells were also precipitated in the upper layer of the sediment (Figure 2D). The cells were suspended in the Williams medium E at a volume five times that of the cell pellet. The number and viability of hepatic parenchymal cells, determined morphologically, were counted on a hemocytometer under 0.08% trypan blue.

### 2.9. Hepatocyte Cryopreservation

Approximately 5 × 10^6^ cells were suspended in 1 mL of the STEM-CELLBANKER and kept in the −80 °C deep freezer for 1 h. Frozen cells were stored in liquid nitrogen until use. When thawing the cells, cryotubes were warmed in a 37 °C warm bath for 90 s until ice clumps disappeared and cell suspension was transferred to 15 mL tubes containing 9 mL of Williams medium E, followed by centrifugation at 190× *g* for 3 min to remove cryopreservant.

### 2.10. Hepatocyte Culture

The 1 × 10^5^ of the parenchymal cells were plated and cultured on type I collagen-coated 6-well dish (354500; Biocoat; Corning) with 2 mL of Williams medium E containing 1 μmol/L dexamethasone, 1 μmol/L insulin, 2 mmol/L GlutaMax (Thermo Fisher Scientific), 10% FBS, and antibiotics (100 mg/mL penicillin G, 100 μg/mL streptomycin, 100 μg/mL kanamycin, and 250 ng/mL amphotericin B; FUJIFILM Wako Pure Chemical Corporation).

### 2.11. Hematoxylin-and-Eosin Staining and Immunostaining of Frozen Sections of Kidney Spheroids

The harvested kidney spheroids were fixed in 4% paraformaldehyde (161-20141; FUJIFILM Wako Pure Chemical Corporation) in PBS at 4 °C overnight and dehydrated in 15% sucrose in PBS overnight and in 30% sucrose in PBS overnight at 4 °C. Specimens were embedded in an OCT compound (Sakura Finetek, Tokyo, Japan), and 5 μm thick frozen sections were prepared. HE staining was performed according to the standard procedures and each sample was examined under the all-in-one fluorescence microscope (BZ-X800; Keyence, Osaka, Japan). For immunostaining, antigen retrieval was performed in HistoVT One (06380-76; Nacalai Tesque, Kyoto, Japan) in a warm bath at 70 °C for 20 min. After blocking with Blocking One Histo (06349-64; Nacalai Tesque) for 10 min at room temperature, the sections were incubated with primary antibodies overnight at 4 °C and then with secondary antibodies conjugated with Alexa Fluor 555 or 647 for 1 h at room temperature. Sections were mounted with Pro-Long Gold Antifade Mountant with 4′,6-diamidino-2-phenylindole (DAPI) (P36931; Thermo Fisher Scientific). Each sample was examined under the confocal microscope (LSM880; Carl Zeiss, Oberkochen, Germany). As primary antibodies, anti-cadherin 6 (CDH6; HPA007456; Atlas Antibodies AB, Bromma, Sweden) for proximal tubules and anti-E-cadherin (ECAD; 610181; Becton, Dickinson and Company, Franklin Lakes, NJ, USA) for distal tubules were used.

## 3. Results

### 3.1. Macroscopic Findings of Fetal Pigs and Their Organs

Fetal MMPs of varying gestational ages were obtained through cesarean section. The fetal body weight increased exponentially with gestational days (Figure 1B,C). On E30, kidneys, hearts, and livers were macroscopically visible although the skin was fragile and easily disintegrated. The fetal organs followed the size order of kidney < heart < liver (Figure 1D).

### 3.2. In Vivo Differentiation Capacity of Cells Harvested from Vitrified and Thawed Fetal Pig Kidneys

The culture condition for mouse fetal kidney spheroids were not appropriated for pig spheroids [11]. Therefore, leveraging insights showing that spheroid maturation can be facilitated in vivo [19,20], we chose to validate the viability of spheroids in vivo. Spheroids prepared from vitrified and thawed kidneys were transplanted under the left renal capsule of NOG mice and recovered after 14 days (Figure 1E). They increased in size (Figure 1E) and exhibited the formation of glomeruli (Figure 1F) and were proximal to distal tubules (Figure 1G), indicating preserved differentiation capacity (Table 1).

### 3.3. Pulsation Capacity of Pig Fetal Heart Cells under Different Cryopreservation Conditions

Porcine fetal hearts were enzymatically processed (Figure 2A) and cultured in a flat dish. Due to their larger sizes compared to the kidneys, the hearts were divided into three parts before undergoing rapid organ freezing through vitrification (Figure 1D). However, pulsating heart muscle cells were not observed in the cells obtained from vitrified hearts (Figure 2B). Furthermore, the cells from hearts that were slowly frozen showed minimal cell adhesion and no pulsating cells either (Figure 2B). Conversely, non-frozen heart cells demonstrated pulsation (Figure 2B). Seeding cells from hearts that were frozen after dissociation resulted in pulsating cells, similar to what was found in the non-frozen hearts (Figure 2B, Table 1, Appendix A).

### 3.4. Development of Direct Tissue Perfusion Method for Porcine Fetal Livers and Evaluation of Obtained Cells under Different Cryopreservation Conditions

Fetal livers weighed 0.064 ± 0.022 g in E30 fetuses (*n* = 45) and 0.114 ± 0.009 g in E35 fetuses (*n* = 4, mean ± SEM, same hereafter) while the percentages of liver weight to body weight were nearly the same at E30 and E35 (6.9 ± 1.8% and 7.0 ± 0.7%, respectively). First, we tried to isolate liver cells through the collagenase digestion of cut liver pieces, but almost no cells were obtained. Thus, we developed the direct tissue perfusion of collagenase using needle puncture (Figure 2C,D, Appendix A). Using this method, we obtained liver cells whose viability and live cell numbers were 90.9 ± 0.8% (*n* = 7) and 3.8 ± 1.5 × 10^7^ cells/gram liver weight (Figure 2F,G). In a preliminary experiment, however, the viabilities and live cell recoveries of cells obtained from livers frozen through rapid freezing or slow freezing in cryopreservant were 35.3% and 8.0 × 10^5^ cells (*n* = 1), and 17.2% and 3.0 × 10^5^ cells, respectively (Figure 2F,G). Parenchymal cells, hepatoblasts at this stage, were distinguished from nonparenchymal cells based on their large sizes, well-defined cell membranes, and dense cytoplasms (Figure 2E). When these cells were frozen and thawed (post-dissociation freezing), the cell survival rate and the number of live liver parenchymal cells were reduced to 58.6 ± 7.3% and 2.1 ± 1.0 × 10^5^ cells, and the recovery rate of live cells was 6.4 ± 3.7% (*n* = 3). On the next day after cell seeding, parenchymal cells began forming colonies in non-frozen cells (Figure 2H). Cryopreserved cells also formed nonparenchymal-cell-dominant colonies (Figure 2I). The results are summarized in Table 1.

## 4. Discussion

The emergence of human embryonic stem cells [21] and human induced pluripotent stem cells [22] has led to a remarkable advancement in human organoid research. Human kidney organoids [23,24,25] and other organoids have been reported. However, it has been pointed out that these organoids are immature compared to fetal or adult human organs [1,2,3]. Additionally, there are cell types for which induction methods have not yet been established.

We propose a novel therapeutic approach called “xenogeneic regenerative medicine,” wherein human NPCs are differentiated on porcine fetal kidney scaffolds to generate human chimeric kidneys. Recently, we reported human and mouse chimeric renal organoids by co-culturing mouse fetal renal progenitor cells with induced human NPCs [11]. Such chimeric organoids comprising human and porcine progenitor cells would serve as valuable tools for studying the differences in developmental processes and inter-species barriers. Additionally, porcine fetal progenitor cells might be able to supply the cell types missed in induced cells and support the growth of human tissues.

In this study, we investigated appropriate methods for the preservation of various organs from valuable porcine fetal specimens to utilize them as cellular sources for forming spheroids or sheet-like structures after single-cell isolation (Table 1). It had been previously reported that vitrified and thawed kidneys could grow and differentiate in vivo [13,14]. This study demonstrated that vitrified (organ rapid freezing) kidneys maintain the in vivo differentiation potential after dissociation and re-aggregation as well. Therefore, validations of the other two freezing methods were not conducted. For hearts, porcine fetal heart cells yielded pulsating heart muscle cells in vitro. The preservation of hearts as whole organs was not suitable even when they were sliced into pieces; however, heart muscle cells remained pulsating after immediate dissociation and freezing (post-dissociation freezing), suggesting the potential for cryopreservation and future use. Regarding livers, we developed a direct tissue perfusion technique to obtain highly viable liver parenchymal cells. Although the cryopreserved cells could be cultured, there were few parenchymal cells, consistent with the commonly observed decrease in cell viability after cryopreservation in hepatocytes [18]. It is required to investigate the extent to which cryopreserved liver parenchymal cells undergo differentiation compared to fresh cells during prolonged culturing.

The reason why organ rapid freezing through vitrification only worked for the kidneys might be attributed, firstly, to the size differences between the organs. Vitrification is routinely used for the cryopreservation of human ovaries but in the form of 1 mm thick slices [16]. Ice crystal formation during freezing can cause cell damage [26], so intercellular fluid should be replaced with highly osmotic cryoprotectants and the tissue needs to be cooled rapidly in liquid nitrogen. However, with larger sizes, insufficient internal penetration of cryoprotectants may cause ice crystal formation. Secondly, the susceptibility to freeze injury may be different among organs. There are no data comparing the susceptibility of adult and fetal organs in pigs or in other animals. However, a report indicates significant variations in the survival rates of frozen rat fetal brain tissue depending on gestational age [27]. Lastly, in this experiment, the collected fetuses were transported to our laboratory, resulting in an ischemic time of approximately 5 h. The differentiation and expansion capacity of mouse fetal kidneys were maintained after refrigeration as whole fetuses, not as isolated organs, for less than 72 h [14]. However, data are lacking regarding other species and organs. In fact, variability in the post-thaw viability of sliced human adult livers has been attributed to the ischemic time before freezing [28]. It is also suggested that the permissible ischemic time limit varies for each organ in adult humans [29]. Porcine fetal hearts and livers may have been damaged in an ischemic environment, leading to varying degrees of damage.

As limitations of this study, there may be room for improvement in freezing methods. Various improvements, including adjustments to the composition of the cryoprotectants and the temperature during each process, have been implemented [30]. It is hoped that continued refinements will lead to the development of vitrification methods that can better preserve cellular function. Additionally, a recently reported preservation method utilizing nanoparticles during vitrification and thawing may be employed as a more suitable organ preservation technique [31]. Cell culture conditions for each organ also require investigation. Exploring appropriate in vitro differentiation methods for porcine fetal organ cells will enable the development of a simpler and more precise evaluation system without the need for transplantation. 

In the future, based on these insights regarding the appropriate processing methods for porcine fetal organ cells, it is anticipated that further analysis will be conducted to explore the differences in organogenesis between pigs and humans, as well as to investigate chimera formation involving human and pig cells.

## 5. Conclusions

As suitable methods for the harvesting and preservation of cells from porcine fetal organs, kidney cells could be preserved through vitrification and heart cells through immediate post-dissociation cell freezing, and liver cells were best utilized in a fresh state obtained through direct tissue perfusion.

## Figures and Tables

**Figure 1 cells-13-00228-f001:**
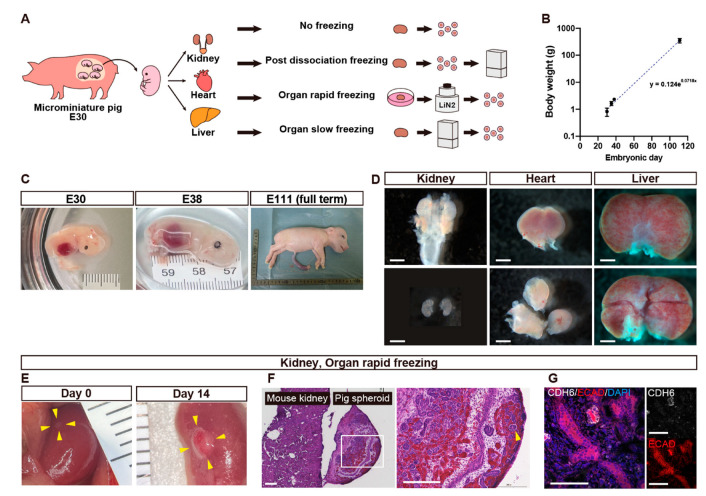
Fetal organ harvesting (**A**–**D**) and quality assessment of cryopreserved kidney cells (**E**–**G**). (**A**) A schematic of the experiment comparing cell viability of organs extracted from microminiature pig (MMP) fetuses among the four cryopreservation conditions. (**B**) A graph showing the relationship between the body weight of fetal MMPs and the embryonic days. Each dot represents a fetus. (**C**) Macroscopic views of fetal MMPs collected on embryonic days 30 (E30), E38, and E111. (**D**) Macroscopic views of kidneys, a heart, and a liver that were extracted from E30 fetal MMPs. (**E**) Macroscopic views of spheroids generated from fetal kidneys that underwent rapid freezing using vitrification, transplanted under the renal capsule of immunodeficient mice on day 0 (left) and day 14 (right). The transplanted spheres have been circled with yellow arrowheads. (**F**) Hematoxylin and Eosin staining of the right panel of (**E**). The white square is enlarged in the right image. The pig spheroid contains glomeruli (arrowheads) and tubules. (**F**) Immunostaining of the right panel of (**E**) showing proximal tubules (CHD6+) and distal tubules (ECAD+). Scale bars: 1 mm in (**D**), 200 μm in (**F**), 100 μm in (**G**). CDH6, cadherin 6; DAPI, 4′,6-diamidino-2-phenylindole; ECAD, E-cadherin.

**Figure 2 cells-13-00228-f002:**
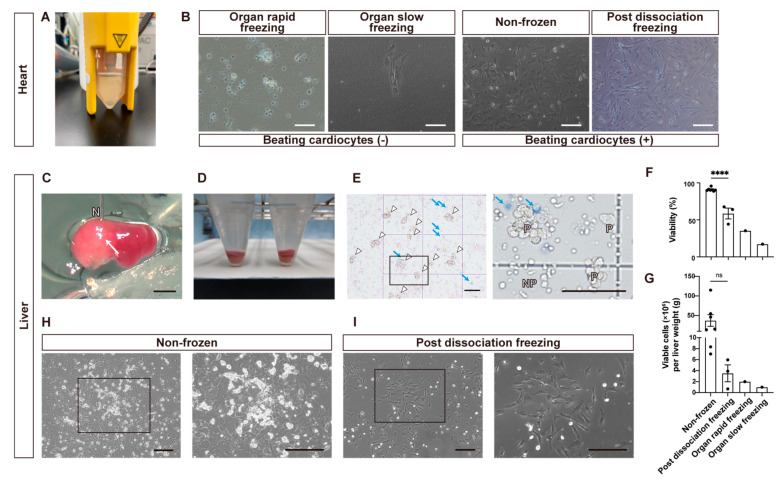
Quality assessment of cryopreserved hearts (**A**,**B**) and livers (**C**–**I**) cells of fetal pigs. (**A**) Cell suspensions of heart cells. (**B**) Microscopic images captured on day 4 after seeding heart cells under four different conditions on a flat surface. Also see Appendix A (**C**) Collagenase digestion of liver parenchyma via microvascular route. The white arrow indicates the lysed area. N: 27-gauge needle. Also see Appendix A. (**D**) Cell pellet precipitated in the bottom of 15 mL tube after centrifugation. (**E**) Microscopic images of isolated non-frozen hepatocytes on a hemocytometer. Live parenchymal hepatocytes are indicated by white arrowheads and dead parenchymal hepatocytes stained with trypan blue are indicated by blue arrows. The black square is enlarged in the right image. P; parenchymal hepatocytes. NP; nonparenchymal cells. (**F**,**G**) Bar graphs comparing the cell survival rate (**F**) and viable cells yielded per gram live weight (**G**) under four conditions: non-frozen, post-dissociation freezing, organ rapid freezing, and organ slow freezing. Each dot represents data obtained from an independent experiment. Ns, not significant; **** *p* < 0.0001. (**H**) Phase-contrast microscope images of primary culture of non-frozen hepatocytes 1 day after seeding. Inside the black square are liver parenchymal cell colonies (enlarged in the right image). (**I**) Phase-contrast microscope images of primary culture of “post dissociation freezing” hepatocytes 1 day after seeding. Inside the black square are nonparenchymal cell colonies, and there are a few parenchymal cells (enlarged in the right image). Scale bars: 200 μm in (**B**,**H**,**I**); 2 mm in (**C**); 100 μm in (**E**).

**Table 1 cells-13-00228-t001:** Comparison of cell viability among cryopreservation conditions in kidneys, hearts, and livers of microminiature pig fetuses.

Condition	Kidney	Heart	Liver
Non-frozen	NA	Beating (+)	Viability was >90% and parenchymal cell colonies were formed.
Post-dissociation freezing	NA	Beating (+)	Viability was around 60% and colonies were nonparenchymal-cell-dominant.
Organ rapid freezing	Differentiation capacity (+)	Beating (−)	Viability was low.
Organ slow freezing	NA	Beating (−)	Viability was low.

## Data Availability

All relevant data supporting the findings of this study are either included within the article or are available upon request from the corresponding author.

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
