# Peer review of "Exploration of Preservation Methods for Utilizing Porcine Fetal-Organ-Derived Cells in Regenerative Medicine Research"

_cells, 2024, doi:10.3390/cells13030228_

Round 1
Reviewer 1 Report
Comments and Suggestions for Authors
Dear Authors,
you present a well-written manuscript how to isolate fetal cells from kidney, liver and heart. However, the title of the manuscript and the abstract are completely misleading for the reader. You do not show any preservation methods for the tissues themselves - instead you only present preservation methods for several cell types. The methods you have chosen for cell isolation are well-known and established in almost all labs. However, the liver tissue perfusion of an embryonic liver is novel and amazing.
What I would have expected considering the title would be novel methods for tissue preservation such as 400um tissue cuts - as used for heart slices - that preserve tissue function and can be frozen. Moreover, distribution of the cryoporotectant before freezing is vital for cell recovery. Please include suggestions for the reader how this can be improved in the tissues - cutting it into pieces is no solution.
Transport conditions of the embryos will influence your cell recovery. Putting all embryos on ice will have negative effects e.g. on heart cell survival.
I think that this paper would still be acceptable for publication by changing the title and rewriting the abstract - pointing out the findings of this manuscript - and adding some advices for the reader.
Author Response
You present a well-written manuscript how to isolate fetal cells from kidney, liver and heart. However, the title of the manuscript and the abstract are completely misleading for the reader. You do not show any preservation methods for the tissues themselves - instead you only present preservation methods for several cell types. The methods you have chosen for cell isolation are well-known and established in almost all labs. However, the liver tissue perfusion of an embryonic liver is novel and amazing.
>Thank you so much for your feedback. We have revised the title and abstract to clearly state that we focused on a preservation method aimed at utilizing fetal organs not as transplantable organs but as a cellular source for research.
What I would have expected considering the title would be novel methods for tissue preservation such as 400um tissue cuts - as used for heart slices - that preserve tissue function and can be frozen. Moreover, distribution of the cryoporotectant before freezing is vital for cell recovery. Please include suggestions for the reader how this can be improved in the tissues - cutting it into pieces is no solution.
>We have included the following statement in the Discussion section as future considerations for tissue cryopreservation methods: “Various improvements, including adjustments to the composition of the cryoprotectants and the temperature during each process, have been implemented [40]. It is hoped that continued refinements will lead to the development of vitrification methods that can better preserve cellular function. Additionally, a recently reported preservation method utilizing nanoparticles during vitrification and thawing may be employed as a more suitable organ preservation technique [41]”.
Transport conditions of the embryos will influence your cell recovery. Putting all embryos on ice will have negative effects e.g. on heart cell survival”.
>As you pointed out, there may be unmeasured impacts of refrigerated transport on fetal organs. We have mentioned this with some modifications in the Discussion section: “The differentiation and expansion capacity of mouse fetal kidneys were maintained after refrigeration as a whole fetus, not as an isolated organ, for less than 72 h [22]. However, the data are lacking regarding other species and organs. In fact, variability in post-thaw viability of sliced human adult livers was attributed to the ischemic time before freezing [38]. It is also suggested that the permissible ischemic time limit varies for each organ in adult humans [39]. Pig fetal hearts and livers may have been damaged in an ischemic environment, leading to varying degrees of damage”.
Reviewer 2 Report
Comments and Suggestions for Authors
The manuscript "Provision of Porcine Fetal Organs as a Stock Source for Research in Regenerative Medicine" is a good attempt in the field of organ xenotransplantation for regenerative medicine. The work seemed conceptualized and executed well and data is also presented well.
However, authors are suggested to incorporate corrections as indicated in the attached edited manuscript.

Author Response
The manuscript "Provision of Porcine Fetal Organs as a Stock Source for Research in Regenerative Medicine" is a good attempt in the field of organ xenotransplantation for regenerative medicine. The work seemed conceptualized and executed well and data is also presented well.
However, authors are suggested to incorporate corrections as indicated in the attached edited manuscript.
>Thank you very much for your feedback on the PDF. We appreciate your insights, and we have made the necessary revisions to the manuscript based on your suggestions.
Round 2
Reviewer 1 Report
Comments and Suggestions for Authors
Dear Authors,
you have adressed all issues and I think the manuscript can be published now.